# Occupational stress and environmental impact among traffic police officers in Kathmandu Valley, Nepal: A qualitative study

**Binita Yadav[1], Sandesh Bhusal[2,3], Anil K. C.[4], Pranil Man Singh Pradhan[5,6]***

**1** Nepal Health Sector Support Program (NHSSP 3), HERD International, Kathmandu, Bagmati, Nepal,
**2** Nepal Health Frontiers, Tokha-5, Kathmandu, Nepal, **3** Department of Allied Health Sciences, University of Connecticut, Storrs, CT, United States of America, **4** Abt Associates, Kathmandu, Bagmati, Nepal, **5** Department of Community Medicine and Public Health, Institute of Medicine, Tribhuvan University, Kathmandu, Nepal, **6** Department of Global Health and Population, Harvard T H Chan School of Public Health, Boston, MA, United States of America

\* pranil.pradhan@gmail.com

**Data Availability Statement:** Data cannot be shared publicly as qualitative data contains personal quotes and clues to where the study occurred and can be potentially identifiable.

## Abstract

Policing is considered an extremely stressful, physically demanding, and mentally challenging occupation. The growing population with an increasing number of vehicles and the harsh working environment has made the work of traffic police even more stressful. This qualitative study aims to examine work and environment-related stressful factors among traffic police officials of Kathmandu Valley, Nepal. This qualitative study used a thematic analysis approach. Face-to-face interviews were conducted with different cadres of 15 traffic police personnel working in Kathmandu Valley, Nepal for at least six months. The study adhered to the Consolidated Criteria for Reporting Qualitative Studies (COREQ) guideline. The findings of this study are structured around five major themes- workload, work-life balance, basic amenities, work environment and occupational health problems, and possible solutions and suggestions. Most of the participants (10 out of 15) reported having work stress due to the heavy workload and hazardous working conditions that had a significant impact on their job performance and mental well-being. This study highlights the challenges faced by traffic police officers in Kathmandu Valley, Nepal. The findings suggest that efforts should be made to improve the working environment of traffic police officers to reduce the physical as well as mental burden among them. The occupational health and safety (OHS) and mental well-being of traffic enforcers' is a critical public health issue, therefore, it should be on the agenda of policymakers, organizational leaders, and stakeholders.

## Introduction

Mental health problems are common in the working population representing a growing concern in the area of occupational health, with potential impacts on workers, organizations, workplace health and compensation authorities, etc. [1]. Law enforcement agencies especially policing have been identified as a hazardous and stressful occupation [2]. Their work has been

Contact for the ethical body is Institutional Review Committee (irc@iom.edu.np) of Institute of Medicine, Tribhuvan University

**Funding:** The authors received no specific funding for this work.

**Competing interests:** The authors have declared that no competing interests exist.

characterized as particularly stressful as their duty is physically and emotionally demanding, as well as lacking flexibility and control [3]. Due to the high-risk nature of frontline duty and the presence of organizational stressors, police officers face an elevated risk of developing psychological illnesses [4].

The prevalence of occupational stress and mental health problems among police officers is a growing concern globally. A study among Portuguese police officers demonstrated that 85% of the sample presented high operational stress levels with 55% of the sample at risk of a psychological disorder [5]. Numerous studies have indicated that in comparison to other professions, law enforcement officers are more susceptible to stress-related physical illnesses such as heart disease, chronic pain, and insomnia, as well as stress-related psychological disorders such as depression, domestic violence, and drug and alcohol abuse [5–7]. Moreover, studies have reported that job stress has consistently increased among police officers in the last decade, and this chronic job stress negatively affects both the person and the organization [5]. At the individual level, job stress can lead to poor mental health, work-family conflict, burnout, suicide, etc. whereas at the organizational level, it affects performance and counterproductive work behaviors [8–10].

Traffic police, a special branch of the police force that serves mostly in traffic or road policing units, face multiple occupational hazards due to the nature of their job. In countries like Nepal, as they have to stand on the road for long hours, they are continuously exposed to vehicular emissions and noisy and polluted environments [11]. These personnel have to undergo physical strain in an environment polluted by fumes, the exhaust of vehicles, the use of blowing horns, the blowing of dust in the air by a speeding vehicle, etc. [12]. Such risky and demanding tasks expose them to difficult and tense scenarios that can affect their mental health and potentially their work quality [13]. Still little has been done to assess their occupational and psychological health status to suggest preventive measures for the upliftment of their health.

Kathmandu Valley, the capital of Nepal, is known for its scenic beauty and cultural heritage. However, rapid urbanization and modernization has led to an increase in environmental pollution, including air pollution, noise pollution, and traffic congestion. These have made traffic officials' duties more stressful [14]. Furthermore, poor road conditions and limited physical infrastructures to facilitate proper traffic functioning have made their work more burdensome [6]. However, there is lack of adequate literature examining the unique stressors and their consequences on mental, occupational, and physical well-being of this particular group of law enforcement officers.

Given the critical role of traffic police officials, it is crucial to understand the psychological strain this puts on officers. Understanding mental health issues and their key risk factors for traffic police forces has immense potential for the wider study of the long-term health of a key front-line service providers [15]. Increasing mental health awareness and education at an organizational level is essential in helping to alleviate the psychological symptoms derived from their occupation [2]. Therefore, the present study aims to examine work and environment-related stressful factors among traffic police officials of Kathmandu Valley, Nepal and inform the policy makers to design effective interventions to contribute on public safety, traffic management, and overall mental well-being and occupational safety of traffic police officials.

## Methods

### Participants

The participants of this study were traffic police personnel working under the different traffic units of Kathmandu Valley, Nepal for at least six months.

Recruitment of participants for the in-depth interview adhered to purposive sampling technique, which entailed inclusion of individuals who were deemed capable of providing the most comprehensive and informative data. This was done by selecting the participants based on nomination done by the traffic unit in charge and by volunteerism. Recruitment of participants was stopped after reaching data saturation.

## Study design and setting

This qualitative study used a thematic analysis approach for an in-depth understanding of the work environment that may cause stress among traffic police personnel. Occupational stress is a context-dependent phenomenon and traffic police officers in Kathmandu Valley face unique stressors related to the challenging traffic conditions, environment factors, and cultural context. A qualitative approach enables a comprehensive understanding of the specific stressors and their nuanced effects. The Consolidated Criteria for Reporting Qualitative Studies (COREQ) guideline was followed to conduct, analyze, and report the findings of this study [16].

We included different cadres of traffic police personnel working in Kathmandu Valley which comprises three big cities of Nepal (Kathmandu, Lalitpur, and Bhaktapur) and employs a large number of traffic police personnel. This study is relevant because traffic police personnel in Kathmandu Valley, which ranks among the most polluted city in the world, face unique mental health challenges as the city undergoes rapid urbanization with a growing number of vehicles on the roads and the severe pollution problem [17].

## Data collection

Before obtaining consent from participants, researchers clearly explained the objectives to the participants, and they were provided an opportunity to ask any questions they may have had. Also, permission was taken to record the interview. All traffic police approached consented to participate in the interview, and there was no refusal to participate.

Two male researchers (SB and AK) were primarily responsible for coordinating and scheduling the interviews and a female researcher (BY) conducted the interviews. The interviews were scheduled as per the preferences of the participants, allowing them to engage in the interview process without any external disturbances. Researchers introduced themselves and established a good rapport with the participants before initiating the interviews. PMSP, a male professor, and an experienced researcher, ensured that the interviews were conducted following COREQ guidelines.

We used an interview checklist that included engagement questions, exploration questions, and exit questions. The interview checklist is included in a supplementary file S1 Text. Participants recruitment and interviews were carried out between the period October 2018 to April 2019 and each interview lasted around 20–30 minutes.

## Data analysis

The audio records and field notes of all interviews were taken in Nepali language which were then transcribed and translated into English language by three researchers (BY, SB, and AK). PMSP reviewed and ensured the quality of translations.

Two researchers (BY and SB) independently read a sample of anonymized transcripts until they became fully immersed in the dataset to develop an analytical framework and identify key themes. Data were analyzed using thematic analysis [18], an approach used for identifying, analyzing, and reporting themes found in qualitative data. The identified themes and categories were then shared with other authors for their review and then finalized by all co-authors.

To provide deeper insights, anonymous verbatim quotations were selected to reflect the genuine viewpoints of the participants.

## Ethical considerations

We obtained ethical approval from the Institutional Review Committee of the Institute of Medicine, Tribhuvan University, Nepal (Reference number: 238 (6-11-E)/075/076). Prior permission was taken from Metropolitan Traffic Police Division, Kathmandu and written informed consent was collected from each participant before the interview. The confidentiality of collected information was strictly maintained and the information was not accessible to others except for the research team. Investigators did not record or share any of the personal information or identity collected during audio recording. Recordings were in laptops secured with passwords. Investigators did not use any identifiable information during the study dissemination and publication process. Data was not shared publicly in this paper as qualitative data contains personal quotes and clues to where the study occurred and can be potentially identifiable.

## Results

A sample of 15 traffic police officers comprising both genders (8 males and 7 females) was selected for participation in the interviews. The selection process was characterized by the inclusion of participants from diverse categories such as police rank, marital status, parental status, etc. Table 1 summarizes the participant demographic information.

The findings of this study are structured around five major themes- workload, work-life balance, basic amenities, work environment and occupational health problems, and possible solutions and suggestions.

## Workload

The majority of the traffic police personnel mentioned having a heavy workload and working in different shifts in a day. The traffic police at the busy stations said that they usually had a

**Table 1. Participants' characteristics.**

| Participant's characteristics | Number |
|---|---|
| **Age group in years** | |
| Mean (Range) | 30.8 (22–46) |
| 20–30 | 10 |
| 31 and above | 5 |
| **Gender** | |
| Male | 8 |
| Female | 7 |
| **Marital status** | |
| Married | 9 |
| Unmarried | 6 |
| **Police rank** | |
| Inspector | 1 |
| Sub-inspector | 2 |
| Assistant sub-inspector | 2 |
| Head constable | 4 |
| Constable | 6 |

duty of 12–15 hours which increases to 16–18 hours on VIPs' travel and other major occasions. Police constables were mentioned to have an even heavier workload as they have to stand in the middle of the roads for extended periods.

*"If we talk about traffic, we have to stand on roads from 5 am in the morning to around 10–11 pm at night. A traffic police normally does a duty of 15 hours and duty hours are quite difficult as compared to civil police." (Male)*

*"In other professions, work can be postponed for tomorrow but in traffic when we reach on road, we should respond immediately." (Female)*

*"During peak hours, higher-ranking officers are deployed to manage traffic and for checking, while constables are on duty for up to 16–18 hours. The flow of traffic is often disrupted by road construction and drinking water issues, which pose challenges for both traffic officers and drivers." (Male)*

### Work-life balance

Even though there was a provision for various types of leaves, the leave was mentioned as a subsidy rather than a right, and participants had difficulty utilizing their whole leave, with leave being canceled during festive seasons, VIP movements, and national-level programs. Eating times were irregular for many, and exercise was difficult to manage due to long shifts. Participants also mentioned having very little time to spend with their families, with some considering coworkers as family. Female participants mentioned difficulties in maintaining a work-life balance, especially after marriage and after having children.

*"Due to the demands of our job, it is challenging to find enough time to spend with our loved ones. We do not get enough time to be with our family. Now my co-workers are like my family." (Male)*

Almost all the female traffic police who had children mentioned that they had breastfed their babies for 2–3 months and after that they started bottle feeding as they did not have long maternity leave.

*"The workload for women often increases after marriage. Before getting married, I had a relatively lighter workload as I used to live in the barracks, which made things easier. However, managing both household responsibilities and work duties has proven to be more challenging after getting married." (Female)*

*"I availed of a 2-month maternity leave and subsequently added an extra month of home leave before promptly resuming my work on the roads. However, this transition is challenging, particularly for individuals who underwent cesarean delivery." (Female)*

### Basic amenities

Many participants mentioned that the equipment and basic amenities provided to them are not enough and timely. However, some participants were happy with the new equipment provided, such as the breathalyzer. Some had water jars available in their traffic bits, however,

using the toilet was challenging, and living arrangements were poor and unhygienic in some barracks.

*"Breathalyzers have proven to be highly effective in detecting cases of drunk driving. Prior to their use, we had to rely on our sense of smell and intuition, which often resulted in disputes over proof of intoxication. However, due to limited equipment availability, the process can sometimes take longer than desired and cause traffic congestion." (Male)*

*"We do not have a proper digital system, we have to work manually with the help of these two hands in the middle of the dirt roads. The basics like traffic lights, zebra cross, cat eye lights, lane, and traffic signals are not enough. There are no signals in Chowk to direct where it is allowed to go and where it is not." (Male)*

*"The living arrangements and working conditions in our location are substandard, with inadequate protective gear provided to work in polluted environments. Our settlements/barracks were damaged during the earthquake, and unfortunately have not yet been repaired, leaving us living in conditions that are not safe or healthy. For instance, in the Maha shaka (Police Division), we are living in structures that resemble animal sheds." (Male)*

*"Simply using a mask is not enough to protect ourselves from the pollution on the roads as we have to blow the whistle in between. Also, the masks we are given are of poor quality and become useless after just a couple of days. In my opinion, using a mask or not using one makes no difference." (Female)*

## Work environment and occupational health problems

Most of the participants mentioned that arguments with vehicle drivers and pedestrians due to traffic rule violations and environmental pollution like dust, smoke, and noise were their main stressors. Also, poor traffic infrastructures and poor technology had led them to be engaged more manually, increasing their job stress.

*"Perhaps some individuals in the valley are unaware of the traffic regulations, however, there are many who intentionally violate the rule, that's a challenge for us to regulate. And we need to argue every day dealing with such people violating the traffic rules which increases our stress. Unfortunately, people do not understand the problem of traffic police, who have to deal with thousands of vehicles and individuals on a daily basis, resulting in increased workload and stress." (Male)*

*"The violation of the traffic rules by the vehicle drivers makes me irritated. The government has not been able to fulfill our basic needs and we are also not able to serve the people and country to the maximum extent." (Male)*

*"The population and vehicles in Kathmandu Valley are constantly rising, yet the condition of the poor narrow roads has not been improved. In addition, the environment is progressively degrading, with an abundance of dust, smoke, and noise pollution, being challenges for us to carry out our duties effectively. Sometimes I feel like the pollution is silently killing us." (Male)*

Few participants mentioned changes in themselves after joining this profession like being angry frequently, sitting lonely, and not paying attention to others.

*"I have noticed a change in myself, nowadays I feel like I am getting angry too soon and easily. When something goes wrong or someone does not comply after I have tried to explain things to them, I find myself becoming agitated. However, I am aware of this issue and actively working to control my emotions." (Female)*

Only a few of the participants mentioned that they had received stress management training and many of them didn't know about such training provided to them. Also, there was no provision for mental health counseling.

*"Once there was a provision of yoga and meditation. But I don't think there is a provision of mental health-related training and counseling sessions." (Male)*

Most of the participants had common health problems like backache, headache, respiratory system problems, gastritis, eye and ear problems, skin problems, etc. And they said that was no provision for scheduled health checkups among traffic police.

*"The current state of pollution in Kathmandu Valley has made it increasingly challenging to work, particularly for us. The environmental quality is worsening, resulting in dirty roads and polluted air that are causing discomfort, such as burning eyes and difficulty in breathing among traffic police officers like us." (Male)*

*"I have been affected with kidney stones. As we have to stand on the roads for long periods, drinking water frequently and urination is not feasible always." (Female)*

Female traffic officers mentioned that they had a difficult time when they are on duty during their periods. One of the participants said,

*"We have more difficultly in changing pads during menstruation, using other's toilet also becomes difficult sometimes. We do not get to change pads for 5–6 hours when we are on duty." (Female)*

## Possible solutions and suggestions

Most of the traffic police officers mentioned increasing traffic awareness to reduce violations, adopting scientific and technology-based equipment and infrastructure, and ensuring sufficient staffing levels will help to reduce stress among them.

*"To improve road safety, it is important to ensure that traffic lights, zebra crossings, stop lines, and centerlines are clearly visible and that there are sufficient traffic signals. However, road division and drinking water division activities, which involve digging the roads, often take a long time to refill, causing additional problems. There should be proper coordination between the traffic police division and other divisions including electricity, road, and drinking water." (Male)*

Similarly, some participants emphasized the importance of promoting public transportation and banning old vehicles. Additionally, a few suggested starting traffic rule awareness campaigns at the school level to promote safe driving habits from an early age."

*"The government should ban vehicles which are older than 20 years or more. Government should promote public vehicles and private vehicles should be less promoted unless the infrastructures are enough to hold them. The roads should be made wider, equipped with traffic signals." (Male)*

Many participants suggested the provision of bonuses and the need for an increase in leave and duty opportunities in their hometowns to enable them to spend time with their families. They also emphasized the importance of regular health checkups to ensure their well-being.

*"There was the provision of bonus given from the collected revenue on road. But it has also been stopped now. It was acting as the positive motivation in the work among traffic." (Male)*

*"Long leaves or deployment in our hometowns would provide us with more time to engage with our family members. Being in close contact with our loved ones can provide us with more energy and motivation to focus on our job." (Female)*

## Discussion

This qualitative study explored the work and environment-related stressful factors among traffic police officials of Kathmandu Valley, Nepal. The major themes that emerged from interviews included: workload, work-life balance, basic amenities, work environment and occupational health problems, and possible solutions/suggestions. The findings highlighted that the traffic police officers in Kathmandu Valley face numerous challenges related to workload and hazardous working conditions which can impact their job performance and mental well-being.

The study found that the heavy workload and long working hours were among the most significant sources of stress among traffic police officers. Unsurprisingly, most participants worked in different shifts in a day, and some reported working up to 16–18 hours, especially during VIPs movements and other national-level occasions. Studies conducted in different parts of the world among police officials reported a significant association of work stressors with psychiatric symptoms. It is frequently observed that police personnel have to exceed their legislative limit on working hours due to the demand of their job. A study in England reported more than one-quarter of police officers typically worked in excess of the legislative limit on working hours and they were significantly more likely to report common mental disorders [19]. Factors like working for long hours, inadequate work schedules, job demands, etc. had major impacts on the mental well-being of the police personnel and have been associated with a higher risk of burnout among them [19–21].

Kathmandu, the capital of the federal government of Nepal, is the most populous city in the country and is one of the fastest-growing metropolitan cities in South Asia. Due to rapid urbanization, there has been a tremendous increase in vehicle numbers, especially private vehicles, in recent years however, neither the road conditions nor the number of traffic police officials have improved to accommodate the change in the environment [6]. These have added more challenges to the job of traffic police in the Kathmandu Valley.

In this study, most of the participants mentioned the problem of not having a proper digital system and having to work manually with the help of two hands in the middle of dirt roads for long hours. The road infrastructure in Kathmandu Valley is often inadequate to handle the increasing number of vehicles. Poorly designed and narrow roads, lack of proper traffic signs and signals, and limited parking spaces add to the challenges faced by traffic police officials

[22, 23]. They may have to spend long hours managing traffic on congested and poorly maintained roads, making their duty hours more difficult and stressful. The workload was reported to be heavier for police constables, who had to stand on roads for extended periods as compared to the senior officials. A similar kind of inverse relationship between rank and working hours is seen in a study conducted among English police officers [19].

Traffic police officers in Kathmandu Valley mentioned difficulties in maintaining a work-life balance. Despite the provision of various types of leaves, the leave was mentioned as a subsidy rather than a right, necessitating the presence of individuals whenever and wherever their duty was assigned. Additionally, most of the interviewed participants had their homes outside the Kathmandu Valley, making it difficult for them to spend time with their families and loved ones leading to minimal emotional and familial support. On the other hand, police officers' indispensable shift work predicts work-family conflict and mediates between police officers' workload and job stress as well as job dissatisfaction [24–27]. Generally, police personnel are considered strong, both mentally and physically, and emotionally stable. Any display of emotional response to work stress can be considered as a sign of weakness or inability to perform their role and goes against being tough, aggressive, and in control [28]. This might be the reason why traffic officers were reluctant to talk about their mental health in the organizational environment.

Police officers' perceived work-related stress does not only result from their work content but also results from their working conditions [29]. Kathmandu Valley experiences a high level of air pollution primarily due to vehicle emissions, dust, and industrial activities and the city has been consistently ranked as one of the world's most polluted cities in recent years [17]. Traffic police spend a considerable amount of time on the dirty roads of Kathmandu often obstructed by activities of road construction, drinking water, and electricity division which involve digging the roads, taking a long time to refill, causing additional problems. Besides mental stress, prolonged exposure to polluted environments has been found to be associated with respiratory problems, eye irritation, and other health issues, affecting the overall well-being and job performance of traffic police officers [22, 23, 30, 31]. Likewise, in our study, similar kinds of health problems were reported by the participants.

The study also showed that arguments with vehicle drivers and pedestrians due to traffic rule violations were the main stressors among the traffic police [32]. Many drivers and pedestrians in Kathmandu Valley do not adhere to traffic rules and regulations. This lack of awareness and discipline adds to the workload and stress of traffic police officials. Educating and enforcing traffic laws, dealing with traffic rule violators, managing conflicts on the roads, etc. were mentioned as mentally and emotionally taxing tasks for traffic officers.

The traffic police officers in this study were also able to recommend some suggestions based on their experience in managing valley traffic. Traffic rule awareness among the public, drivers, and pedestrians, along with including traffic rule education in the school curriculum, making traffic systems more scientific and technology-based with well-equipped infrastructures, etc. were among the major suggestions provided by the participants. Also, participants highlighted the need for the provision of bonuses and the need for an increase in leave and duty opportunities in their hometowns to enable them to spend time with their families. They also emphasized the importance of regular health checkups to ensure their well-being.

## Strengths and limitations

There were several limitations in this study. The findings of the study are specific to the unique context of Kathmandu Valley and may not be directly applicable to other regions or countries. Also, cultural, social, and infrastructural characteristics specific to the valley may limit the

transferability of the findings. Even though we tried our best to conduct the interviews at a convenient time for the police officers, such an aim may not have been achieved in all cases considering their hectic duty schedule in the Valley.

Despite the limitations, this study explored the occupational stress and working environment among traffic police in Kathmandu Valley, Nepal through in-depth interviews. The findings could be useful to the government, police departments, and other concerned authorities in designing interventions to alleviate the problems and improve the mental well-being of traffic officers.

## Conclusion

The study highlights the challenges faced by traffic police officers in Kathmandu Valley, Nepal, related to occupational stress and harsh working environment. The findings suggest that efforts should be made to improve the working conditions of traffic police officers, such as providing sufficient leave and flexible schedules to have a better work-life balance, better amenities, and technology to aid their work and reduce the burden. There is a need to improve the traffic infrastructure and control the pollution caused by vehicles to reduce the environmental impact on the officers. Also, periodic health screening, provision of stress management training, and mental health counseling can help reduce the psychological stress and early identification of any health problems among traffic police in such a physically and mentally challenging environment of Kathmandu Valley. The occupational health and safety (OHS) and mental well-being of traffic enforcers' is a critical public health issue, therefore, it should be on the agenda of policymakers, organizational leaders, and stakeholders.

## Supporting information

**S1 Text. Interview guideline.**
(DOCX)

**S2 Text. Inclusivity in global research questionnaire.**
(DOCX)

**S1 Checklist. Consolidated Criteria for Reporting Qualitative research checklist.**
(PDF)

## Author Contributions

**Conceptualization:** Binita Yadav, Pranil Man Singh Pradhan.

**Data curation:** Binita Yadav, Anil K. C.

**Formal analysis:** Binita Yadav, Sandesh Bhusal, Anil K. C., Pranil Man Singh Pradhan.

**Investigation:** Binita Yadav.

**Methodology:** Binita Yadav, Sandesh Bhusal, Anil K. C., Pranil Man Singh Pradhan.

**Project administration:** Binita Yadav.

**Resources:** Binita Yadav.

**Software:** Binita Yadav, Sandesh Bhusal.

**Supervision:** Anil K. C., Pranil Man Singh Pradhan.

**Validation:** Binita Yadav, Sandesh Bhusal, Anil K. C., Pranil Man Singh Pradhan.

**Writing – original draft:** Binita Yadav, Sandesh Bhusal, Pranil Man Singh Pradhan.

**Writing – review & editing:** Binita Yadav, Sandesh Bhusal, Anil K. C., Pranil Man Singh Pradhan.

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
