## [Decision Letter · Decision Letter 0]

7 Aug 2023

PGPH-D-23-00984

Occupational stress and environmental impact among traffic police officers in Kathmandu valley, Nepal: a qualitative study

Dear Dr. Pradhan,

Thank you for submitting your manuscript to PLOS Global Public Health. After careful consideration, we feel that it has merit but does not fully meet PLOS Global Public Health’s publication criteria as it currently stands. Therefore, we invite you to submit a revised version of the manuscript that addresses the points raised during the review process.

We look forward to receiving your revised manuscript.

Kind regards,

Prof Razak M Gyasi, PhD, PD

Academic Editor

Journal Requirements:

2. Please include a complete copy of PLOS’ questionnaire on inclusivity in global research in your revised manuscript. Our policy for research in this area aims to improve transparency in the reporting of research performed outside of researchers’ own country or community. The policy applies to researchers who have travelled to a different country to conduct research, research with Indigenous populations or their lands, and research on cultural artefacts. The questionnaire can also be requested at the journal’s discretion for any other submissions, even if these conditions are not met.  Please find more information on the policy and a link to download a blank copy of the questionnaire here: https://journals.plos.org/globalpublichealth/s/best-practices-in-research-reporting. Please upload a completed version of your questionnaire as Supporting Information when you resubmit your manuscript.

Additional Editor Comments:

Dear Authors,

Thank you again for submitting your manuscript to this journal. The reviewers have raised very important concerns that require very critical attention, particularly about the sample size the procedures in the selection of the same. Many other comments are attached in the draft manuscript for execution. Please, address these comments and edits carefully to ensure the progress of your paper. Very importantly, highlight clearly any changes to the manuscript during the revision to allow reassessment of the manuscript. The manuscript will be returned if this is not done appropriately.

Reviewers' comments:

Reviewer's Responses to Questions

**Comments to the Author**

1. Does this manuscript meet PLOS Global Public Health’s publication criteria? Is the manuscript technically sound, and do the data support the conclusions? The manuscript must describe methodologically and ethically rigorous research with conclusions that are appropriately drawn based on the data presented.

Reviewer #1: Yes

Reviewer #2: Yes

Reviewer #3: Yes

Reviewer #4: Yes

2. Has the statistical analysis been performed appropriately and rigorously?

Reviewer #1: N/A

Reviewer #2: N/A

Reviewer #3: N/A

Reviewer #4: No

3. Have the authors made all data underlying the findings in their manuscript fully available (please refer to the Data Availability Statement at the start of the manuscript PDF file)?

Reviewer #1: Yes

Reviewer #2: No

Reviewer #3: Yes

Reviewer #4: Yes

4. Is the manuscript presented in an intelligible fashion and written in standard English?

Reviewer #1: Yes

Reviewer #2: Yes

Reviewer #3: Yes

Reviewer #4: Yes

5. Review Comments to the Author

Reviewer #1: Dear Authors,

Your work is well put together.

These are a few things you should take a look at.

Abstract:

Lines 29-30: The six themes that you identified may perhaps be presented more distinctively, especially the "work environment and health problems".

Suggestions:

1. Since you referred to "occupational health" in Line 362 of your conclusion, I suggest you modify the "work environment and health problems" into "Work environment and occupational health problems" since a work environment may not necessarily translate into health problems.

Line 31: While I understand that your research is qualitative, I still suggest you support the work "Most" with numerical data, like how many participants reported this. A simple number or percentage will give a clearer picture.

Introduction:

Lines 62-64: Perhaps replace this sentence "These types of hazardous roles and responsibilities place them in challenging and stressful situations that can significantly impact their mental well-being and possibly even performance" with this

“Such risky and demanding tasks expose them to difficult and tense scenarios that can affect their mental health and potentially their work quality.” for more clarity.

Methods:

Lines 99-102: This expression "The study of mental health among traffic police personnel in Kathmandu Valley is particularly relevant as the city's rapid urbanization has resulted in an increase in the number of vehicles on the roads and the issue of pollution is also of great concern, with the city being ranked as one of the most polluted cities in the world", may be replaced with "This study is relevant because traffic police personnel in Kathmandu Valley, which ranks among the most polluted city in the world, face unique mental health challenges as the city undergoes rapid urbanization with a growing number of vehicles on the roads and the severe pollution problem."

Lines 116: "Participants were recruited and interviews were carried out between the period October 2018 to..." should be corrected to "Participants recruitment and interviews were carried out between the period October 2018 to..."

Results:

Lines 139-140: Please refer to comments on lines 29-30 above.

Line 240: Check the spelling of "menstruation". It is spelt here as "mensuration", I am not sure if you wanted to leave it as pronounced by participants.

References:

Please add the DOIs or URLs of all the references listed in the document

Line 439: Reference 27 has some boxes in the lines, kindly fix them

Reviewer #2: The manuscript does not explicitly mention whether the authors have made all data underlying the findings available. As a qualitative study, data availability may vary depending on the specific research context and ethical considerations. In this particular case, qualitative data may be sensitive and require careful anonymization or protection to ensure participants' confidentiality. i think it is appropriate for the authors to provide clarity on data availability or justification for not sharing the data in this manuscript.

Reviewer #3: Based on the defined scope of this study, the authors have adequately demonstrated that understanding occupational stress and environmental impact among traffic police officers can be one way to improving the quality of their work life, enhancing public safety, and creating a healthier and more resilient workforce.

The qualitative manuscript has explored the experiences and coping mechanisms of traffic police officers in a busy metropolitan area, focusing on occupational stress and its impact on their well-being. Through in-depth interviews and thematic analysis, the study has revealed the significant challenges officers face, and emphasized the need for organizational support, including mental health resources and flexible work schedules, to enhance well-being. Overall, the research sheds light on the complex interplay between occupational stress and environmental factors, offering valuable insights for policy development and fostering a healthier and more effective traffic policing environment.

You can find my minor edits in the manuscript attached.

Reviewer #4: Few minor questions which the author need to answer

1. How the study is different from 'Prevalence and factors associated with symptoms of depression, anxiety and stress among traffic police officers in Kathmandu, Nepal: a cross-sectional survey'. plus there are some other similar studies carried out in the same area. give a proper justification.

2. How the sample size of 15 is chosen?

3. there is statistical analysis shown in the manuscript.

6. PLOS authors have the option to publish the peer review history of their article (what does this mean?). If published, this will include your full peer review and any attached files.

**Do you want your identity to be public for this peer review?** For information about this choice, including consent withdrawal, please see our Privacy Policy.

Reviewer #1: **Yes: **Blessing O. Josiah

Reviewer #2: **Yes: **Dr Ivan Namakoola

Reviewer #3: No

Reviewer #4: No

---

## [Decision Letter · Decision Letter 1]

30 Oct 2023

Occupational stress and environmental impact among traffic police officers in Kathmandu valley, Nepal: a qualitative study

PGPH-D-23-00984R1

Dear Dr. Pradhan,

We are pleased to inform you that your manuscript 'Occupational stress and environmental impact among traffic police officers in Kathmandu valley, Nepal: a qualitative study' has been provisionally accepted for publication in PLOS Global Public Health.

Best regards,

Razak M Gyasi, PhD, PD

Academic Editor

Reviewer Comments (if any, and for reference):

Reviewer's Responses to Questions

**Comments to the Author**

1. If the authors have adequately addressed your comments raised in a previous round of review and you feel that this manuscript is now acceptable for publication, you may indicate that here to bypass the “Comments to the Author” section, enter your conflict of interest statement in the “Confidential to Editor” section, and submit your "Accept" recommendation.

Reviewer #2: All comments have been addressed

Reviewer #3: All comments have been addressed

2. Does this manuscript meet PLOS Global Public Health’s publication criteria? Is the manuscript technically sound, and do the data support the conclusions? The manuscript must describe methodologically and ethically rigorous research with conclusions that are appropriately drawn based on the data presented.

Reviewer #2: Yes

Reviewer #3: Yes

3. Has the statistical analysis been performed appropriately and rigorously?

Reviewer #2: N/A

Reviewer #3: N/A

4. Have the authors made all data underlying the findings in their manuscript fully available (please refer to the Data Availability Statement at the start of the manuscript PDF file)?

Reviewer #2: Yes

Reviewer #3: Yes

5. Is the manuscript presented in an intelligible fashion and written in standard English?

Reviewer #2: Yes

Reviewer #3: Yes

6. Review Comments to the Author

Reviewer #2: I am satisfied with the responses that the authors have done to improve the manuscript.

Reviewer #3: In my view, this manuscript is now in good shape. Thank you.

7. PLOS authors have the option to publish the peer review history of their article (what does this mean?). If published, this will include your full peer review and any attached files.

**Do you want your identity to be public for this peer review?** For information about this choice, including consent withdrawal, please see our Privacy Policy.

Reviewer #2: **Yes: **Dr Ivan Namakoola

Reviewer #3: No
